# First unambiguous record of pneumaticity in the axial skeleton of alvarezsaurians (Theropoda: Coelurosauria)

**G.J. Windholz**[1]*, **J.G. Meso**[1]☉, **M.J. Wedel**[2]☉, **M. Pittman**[3]*

**1** CONICET- Instituto de Investigación en Paleobiología y Geología, Sede Alto Valle-Valle Medio Universidad Nacional de Río Negro, General Roca, Río Negro, Argentina, **2** College of Osteopathic Medicine of the Pacific and College of Podiatric Medicine, Western University of Health Sciences, Pomona, California, United States of America, **3** School of Life Sciences, The Chinese University of Hong Kong, Shatin, Hong Kong SAR, China

☉ These authors contributed equally to this work.
* gwindholz@unrn.edu.ar (GJW); mpittman@cuhk.edu.hk (MP)

## Abstract

*Bonapartenykus ultimus* is an alvarezsaurid theropod from the Upper Cretaceous of Patagonia, Argentina. This species is represented by the holotype specimen and several referred specimens, many of which have pneumatic structures. Pneumaticity involves the invasion of the interior of the skeleton by means of air sac diverticula. Such invasion occurs from cortical openings (foramina) that communicate with internal air spaces. Despite previous studies on pneumatic structures in theropod specimens, there are no studies focusing on the family Alvarezsauridae. Here we address this gap by presenting the first contribution focusing on the pneumatic features of an alvarezsaurid theropod using both external skeletal anatomy as well as computed tomographic images showing internal details. The specimens studied show that the axial skeleton of *B. ultimus* was invaded by pneumatic structures, reaching the middle section of the tail. Our study suggests that pneumaticity among alvarezsaurids did not have a linear evolutionary trajectory, but instead shows a more random pattern of variability. This study is an important first step that paves the way for future studies to uncover the extent of pneumatic invasion among alvarezsaurids and its macroevolutionary implications.

## Introduction

Alvarezsauridae are a group of early diverging maniraptoran theropod dinosaurs, first recognized by Bonaparte [1]. The fossil record of this dinosaur family comes from the Upper Cretaceous of South America, North America, Europe, and Asia [2–11]. Notably, the diversity of this clade in South America is limited to Argentina where five species have been described to date: *Alnashetri cerropoliciensis* from the Cenomanian Candeleros Formation [7,12]; *Patagonykus puertai* from the Coniacian Portezuelo Formation [3,13]; *Alvarezsaurus calvoi* and *Achillesaurus manazzonei* from the Coniacian Bajo de la Carpa Formation [1,4]; and *Bonapartenykus ultimus* from middle Campanian–lower Maastrichtian Allen Formation [6].

**Data availability statement:** All relevant data are within the manuscript.

**Funding:** We thank P. Chafrat from Museo Patagónico de Ciencias Naturales, General Roca, Río Negro Province, Argentina. The authors gratefully acknowledge "Fundacion Patagonica de Ciencias Naturales" and "Sanatorio Juan XXIII" for making the CT images possible. MP was supported by the Faculty of Science of The Chinese University of Hong Kong. We thank Hans-Dieter Sues, an anonymous reviewer, and the editorial team of PLOS ONE for their comments which improved the quality of this manuscript.

**Competing interests:** The authors have declared that no competing interests exist.

Among the Argentinian forms, an endemic subclade Patagonykinae has been identified that currently includes *Patagonykus puertai* and *Bonapartenykus ultimus* [6]. This subfamily shows a tendency towards larger body sizes relative to other alvarezsaurians with body length estimated between ~ 3 to ~ 3.3 meters [14]. *Bonapartenykus ultimus* comes from the "Salitral Ojo de Agua" locality of Rio Negro Province (Patagonia, Argentina) and is the only known species of its genus. The species is represented by the holotype specimen published by Agnolín et al. [6], along with several referred specimens published by Salgado et al. [15] and Meso et al. [16]. These specimens exhibit particular bone anatomy including structures indicating the presence of postcranial pneumaticity.

Pneumaticity is a widely distributed feature among archosaurs, including pterosaurs and saurischian dinosaurs [17–24]. However, birds are the only living archosaurs with postcranial pneumaticity in their skeleton, since crocodylians lack it [23]. The pneumatic system denotes the invasion of bone by air sac diverticula, which may manifest through observable external openings (i.e., fossae and foramina) and internal pneumatization patterns (e.g., procamerate, camerate and camellate). Numerous studies have explored the pneumatic system of dinosaurs [e.g., 20,21,25–28]. Sauropods have been the main focus of attention [e.g., 19–21,29,30], but there have also been studies investigating pneumaticity in non-avian theropods [22,23,28,31,32,34,35]. Nevertheless, to date, no studies have focused on the pneumaticity of the family Alvarezsauridae. This study aims to fill this knowledge gap by characterizing the pneumatic structures in the axial skeleton of *Bonapartenykus* based on first-hand study of external anatomy as well as internal anatomy from computed tomography scans. The observed pneumatic pattern was compared with other theropods, focusing on alvarezsaurians, to understand its broader significance in theropod evolution.

## Materials and methods

The fossils described here come from the middle Campanian-lower Maastrichtian (Upper Cretaceous) Allen Formation of Río Negro Province, Argentina. The holotype specimen of *Bonapartenykus* (MPCA 1290) is housed in the MPCA "Museo Provincial Carlos Ameghino", Cipolletti City. Referred specimens (MPCN-PV 738) are housed in the MPCN "Museo Patagonico de Ciencias Naturales" General Roca. Both museums are located in Río Negro Province, Argentina. No permits were required for the described study, which complied with all relevant regulations.

Computed tomographic images (CTs) were obtained from eleven vertebral elements: a mid-cervical vertebra (MPCN-Pv 738.28), two posterior cervical vertebrae (MPCN-Pv 738.26 and 27), two cervico-dorsal vertebrae (MPCN-Pv 738.15 and 16), a sacral vertebra (MPCN-Pv 738.14), the first caudal vertebra (MPCN-Pv 738.10), an anterior caudal vertebrae (MPCN-Pv 738.29), two mid-caudal (MPCN-Pv 738.8 and 30) and a posterior caudal vertebra (MPCN-Pv 738.11). The scans were performed at "Sanatorio Juan XXIII" hospital in General Roca, Río Negro Province, Argentina. The remaining elements were examined first-hand for external anatomical features and to a limited extent some internal anatomy accessible through natural fractures in the fossils. The descriptions follow the vertebral laminae and fossae nomenclature proposed by Wilson [36,37] and Wilson et al. [38] as well as the terminology for pneumatic structures of Britt [17], Wedel [20], and O'Connor [22].

## Results

Two anterior cervical vertebrae assigned to cf. *Bonapartenykus ultimus* (MPCN-PV 738) are preserved, a vertebral centrum (MPCN-Pv 738.32) and an incomplete neural arch (MPCN-Pv 738.47). The external surface of anterior cervical centrum (MPCN-Pv 738.32) lacks foramina,

but has a deep depression on its ventral region. This centrum is broken dorsally, exposing a wide internal cavity surrounded by a reticule of interconnected trabeculae (Fig. 1A). The anterior cervical neural arch (MPCN-Pv 738.47) bears poor developed laminae and shallow neural fossae (Fig. 1B) as well as a pair of foramina on the inner surface of centrodiapophyseal fossa (**cdf**).

The mid-cervical region of cf. *Bonapartenykus ultimus* is represented by a neural arch (MPCN-Pv 738.28). This has a rough ventral surface indicating an open neurocentral joint (i.e., site of attachment with the centrum). This element presents a pattern of poorly developed neural laminae, accompanied by wide lateral fossae (i.e., **cdf**, **prcdf**, **pocdf** and **sdf**) (Fig. 2A). Additionally, natural fractures in the preserved transverse processes and base of the neural spine reveal a three-dimensional network of trabeculae, with small interconnected air spaces. In particular, the inner surface of the prezygapophyseal centrodiapophyseal fossa (**prcdf**) and spinodiapophyseal fossa (**sdf**) of MPCN-Pv 738.28 bears foramina (Fig. 2A).

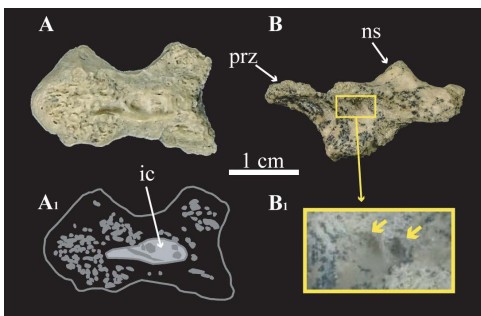

**Fig 1. Anterior cervical vertebrae assigned to cf. *Bonapartenykus ultimus*.** A, MPCN-Pv 738.32 in dorsal view; A₁, interpretive drawing showing internal air spaces; B, MPCN-Pv 738.47 in lateral view; B₁, detail of foramina. Abbreviations: **ic**, internal cavity; **ns**, neural spine; **prz**, prezygapophysis.

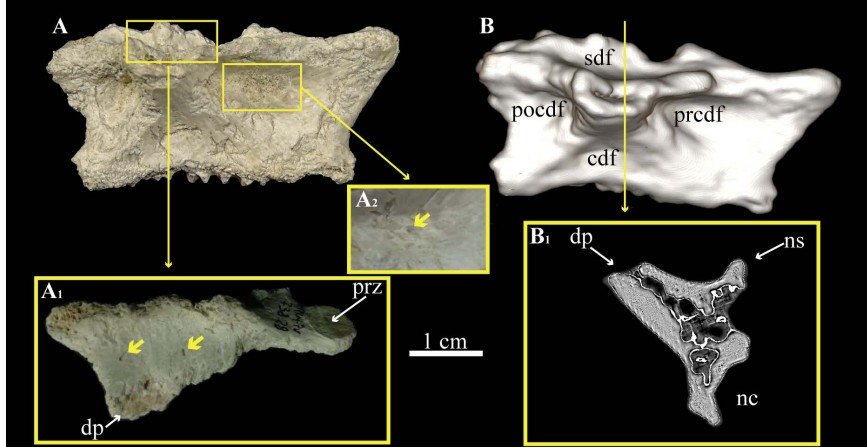

**Fig 2. Mid-cervical vertebrae MPCN-Pv 738.28 assigned to cf. *Bonapartenykus ultimus*.** A, MPCN-Pv 738.28 in lateral view; A₁, close ups showing foramina on **sdf** and A₂, on **prcdf**; B, three-dimensional reconstruction of MPCN-Pv 738.28 in lateral view; B₁, transverse section of MPCN-Pv 738.28 at mid-length, note the internal air spaces. Abbreviations: **cdf**, centrodiapophyseal fossa; **nc**, neural canal; **ns**, neural spine; **dp**, diapophysis; **prcdf**, prezygapophyseal centrodiapophyseal fossa; **prz**, prezygapophysis; **pocdf**, postzygapophyseal centrodiapophyseal fossa; **sdf**, spinodiapophyseal fossa.

Although mid-cervical neural fossae are shallow externally (Fig. 2B), CT scan images reveal a network of internally interconnected air spaces (Fig. 2B).

The posterior cervical vertebrae assigned to cf. *Bonapartenykus ultimus* (MPCN-Pv 738.26 and 27) preserve only the neural arches. The pattern of neural laminae and fossae is considerably more complex compared to the mid-cervical vertebrae (Fig. 3A and B). Lateral fossae (i.e., **cdf**, **prcdf**, **pocdf** and **sdf**) are deep and almost reach the axial plane of the element. The posterior cervical vertebra MPCN-Pv 738.27 has a deep spinopostzygapophyseal fossa (**spof**) with an elliptical outline (Fig. 3C). This fossa reaches almost to the roof of the neural canal, invading almost half the height of the neural arch. The **spof** is not discernible in posterior cervical vertebra MPCN-Pv 738.26 since the posterior surface of the neural spine is completely eroded. The inner surfaces of the prezygapophyseal centrodiapophyseal fossae (**prcdf**) have a pair of foramina in MPCN-Pv 738.26 and a simple foramen in MPCN-Pv 738.27. Additionally, both posterior cervical vertebrae have a poorly developed lamina on the inner surface of the spinodiapophyseal fossae (**sdf**) and have a foramen on each side of that lamina; the only exception is the right **sdf** of MPCN-Pv 738.26, which is simple and has only one foramen inside it. Interestingly, CT scans also show the interior of the neural arch with interconnected air spaces, including some small camerae in the neural spine (Fig. 3D and E).

The cervico-dorsal transition of cf. *Bonapartenykus ultimus* is represented by two incomplete vertebrae, with both MPCN-Pv 738.15 and 16 lacking the centrum, as well as an isolated fragment of postzygapophysis (MPCN-Pv 738.48) and neural spine (MPCN-Pv 738.49). Their neural fossae (**prsdf**, **posdf**, **sprf** and **spof**) are wide and deep (Fig. 4A-D), especially the postzygapophyseal spinodiapophyseal fossa (**posdf**) that invades almost to the axial plane of the neural arch (Fig. 4E and F). The cervico-dorsal transition is characterized by the presence

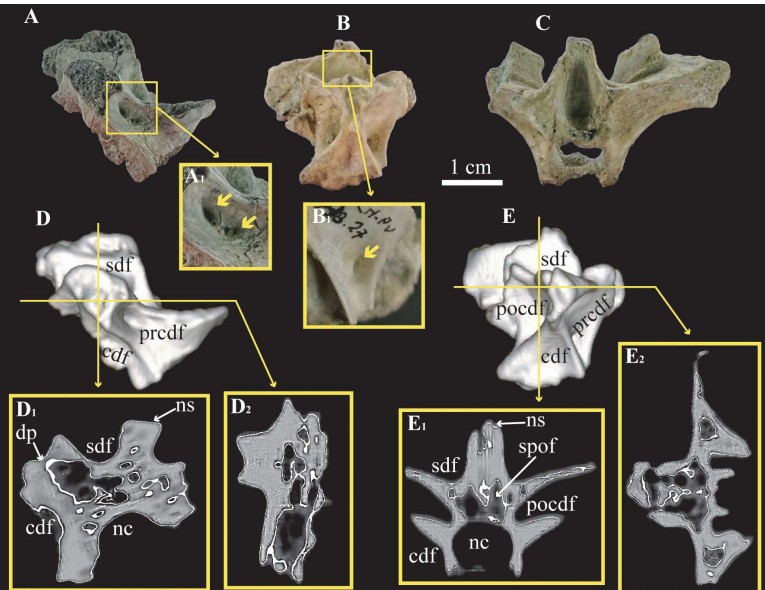

**Fig 3. Posterior cervical vertebrae assigned to cf. *Bonapartenykus ultimus*.** A, MPCN-Pv 738.26 in lateral view, and detail of foramina (A₁); B and C, MPCN-Pv 738.27 in lateral view (B), detail of foramina (B₁), and posterior views (C); D, three-dimensional reconstruction of MPCN-Pv 738.26 in lateral view, transverse (D₁), and frontal (D₂) sections; E, three-dimensional reconstruction of MPCN-Pv 738.27 in lateral view, transverse (E₁), and frontal (E₂) sections. Abbreviations: **cdf,** centrodiapophyseal fossa; **dp**, diapophysis; **nc**, neural canal; **ns**, neural spine; **prcdf**, prezygapophyseal centrodiapophyseal fossa; **pocdf**, postzygapophyseal centrodiapophyseal fossa; **sdf** spinodiapophyseal fossa; **spof**, spinopostzygapophyseal fossa.

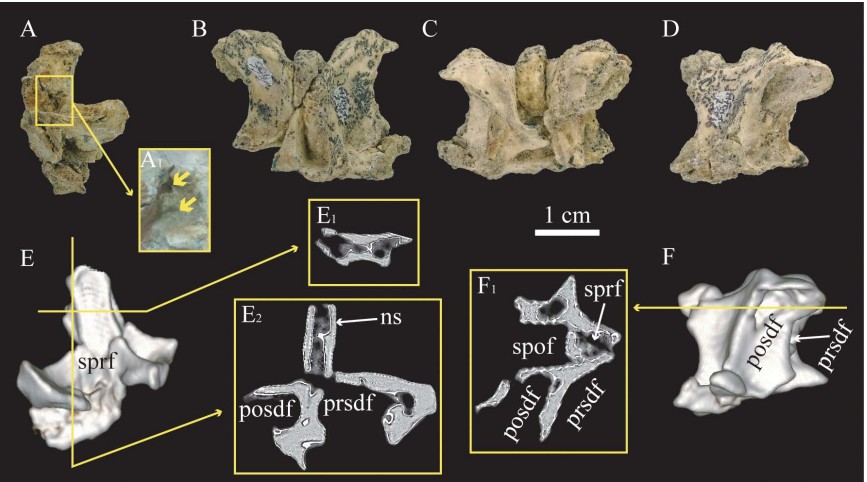

**Fig 4. Cervico-dorsal vertebrae assigned to cf. *Bonapartenykus ultimus.*** A, MPCN-Pv 738.15 in lateral view, and detail of two foramina (A₁); B, C, and D, MPCN-Pv 738.16 in anterior (B), posterior (C), and lateral views (C); E, three-dimensional reconstruction of MPCN-Pv 738.15 in anterior view, frontal (E₁), and parasagittal (E₂) sections; F, three-dimensional reconstruction of MPCN-Pv 738.16 in lateral view and frontal section (F₁). Abbreviations: **ns**, neural spine; **posdf**, postzygapophyseal spinodiapophyseal fossa; **prsdf**, prezygapophyseal spinodiapophyseal fossa; **spof**, spinopostzygapophyseal fossa; **sprf**, spinoprezygapophyseal fossa.

of deep spinoprezygapophyseal (**sprf**) and spinopostzygapophyseal (**spof**) fossae with an elliptical contour. This anatomical area is also represented by a fragment of isolated postzyga-pophysis (MPCN-Pv 738.48) and neural spine (MPCN-Pv 738.49), whose fractures show large internal camerae. The incomplete vertebra MPCN-Pv 738.15 has two foramina, separated by a thin bony layer on the interior surface of prezygapophyseal spinodiapophyseal fossa (**prsdf**). CT images show that the interior of the neural spine of MPCN-Pv 738.15 and 16 has abundant interconnected air spaces ([Fig. 4E] and [F]).

The mid-dorsal vertebra (MPCA 1290, holotype specimen) is almost completely preserved and its anatomy has been described in detail by Agnolín et al. [6]. However, it is interesting to note some features that could be linked to the presence/absence of pneumaticity in the only preserved vertebral element of the holotype specimen of *Bonapartenykus ultimus*. For example, the centrum lacks pleurocoels, although it bears a foramen on its lateral cortical surface. The natural fractures in the centrum do not appear to show signs of internal pneumaticity, such as camellae or camerae, although the preservation in that area is poor. The neural arch has well-developed lamellae that delimit wide and deep **pacdf**, **pocdf** and **sdf**; and a much-reduced **prcdf** (Fig. 5A-B). The neural arch also has a deep **spof** ([Fig. 5C]).

The sacral region of cf. *Bonapartenykus ultimus* is represented by two anterior sacral ver-tebrae (MPCN-Pv 738.14) and the last sacral vertebra (MPCN-Pv 738.31). The poor state of preservation of these elements makes it difficult to observe certain features; however, natural fractures in their centra show the presence of small camerae internally (Fig. 6A-D). CT scan images show a large cavity in the posterior region of the centrum that occupies almost the entire interior and is surrounded by small air spaces. The anterior half of centrum is com-posed of cancellous bone ([Fig. 6E]).

The exterior of the anterior and middle caudal vertebrae of cf. *Bonapartenykus ultimus* (MPCN-Pv 738.8, 10, 29, 30, 37 and 38) show lateral depressions and rather shallow neural fossae. The lateral surfaces of some caudal centra bear small and elliptical foramina, and natu-ral fractures in the centra reveal internal air-spaces. CT images show that the interior of these

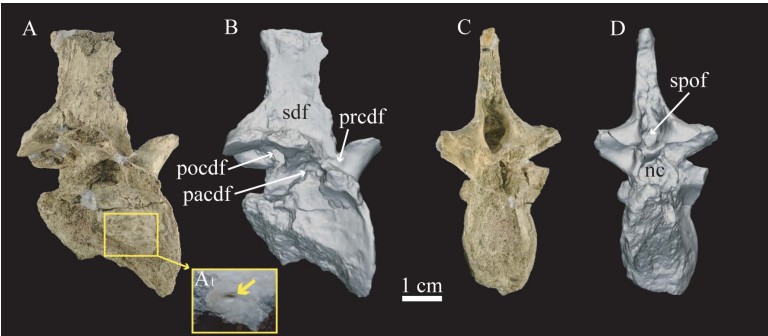

**Fig 5. Mid-dorsal vertebrae of holotype specimen of *Bonapartenykus ultimus* (MPCA 1290).** A, vertebra in lateral view and detail of foramina (A₁); B, three-dimensional reconstruction in lateral view; C, vertebral element in posterior view; D three-dimensional reconstruction in posterior view. Abbreviations: **nc**, neural canal; **pacdf**, parapophyseal centrodiapophyseal fossa; **pocdf**, postzygapophyseal centrodiapophyseal fossa; **prcdf**, prezygapophyseal centrodia-pophyseal fossa; **sdf** spinodiapophyseal fossa; **spof**, spinopostzygapophyseal fossa.

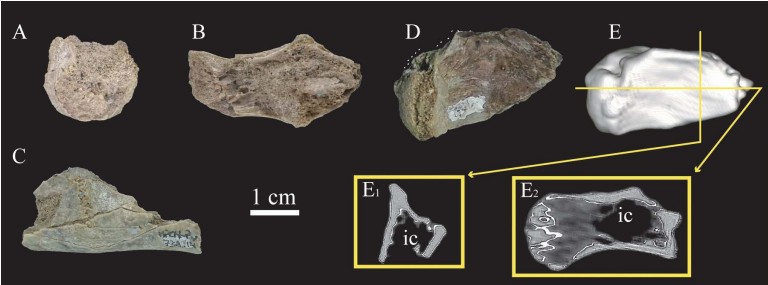

**Fig 6. Sacral vertebrae assigned to cf. *Bonapartenykus ultimus*.** A, B and C, MPCN-Pv 738.14 in anterior (A), dorsal (B) and lateral (C) views; D, MPCN-Pv 738.31 in lateral view; E, three-dimensional reconstruction of MPCN-Pv 738.31 in lateral view, and transverse (E₁), and frontal (E₂) sections. Abbreviation: **ic**, internal cavity.

vertebrae are mainly invaded by asymmetrical air-spaces, however, this feature is variable in the vertebrae scanned. For instance, the first caudal centrum (MPCN-Pv 738.10) is pneumatic with camellate tissue, except for the condyle which is solid (Fig. 7A-B). In contrast, a big camera is present in anterior caudal centrum MPCN-Pv 738.29, with a longitudinal septum in its dorsal area (Fig. 7C-D). In the mid-caudal vertebrae (MPCN-Pv 738.30 and MPCN-Pv 738.8), CT scans show cancellous bone in the interior of the centrum, including the condyle (Fig. 7E-I). Remarkably, the most complete caudal vertebra MPCN-Pv 738.8 has a large internal air-chamber in the anterior half of the neural arch, above the neural canal, which bifurcates in the posterior half of the neural arch. This air-structure contacts an external foramen located at the base of the neural spine (Fig. 7I) and pneumatizes almost the entire neural arch. Finally, the posterior caudal centrum MPCN-Pv 738.11 has a foramen on its lateral surface, but CT images show that its interior is completely solid (Fig. 7J-K).

## Discussion

Among theropods, alvarezsaurids exhibit a distinctive suite of skeletal characteristics [14]. Members of this clade were characterized by a gracile skull with small teeth, that likely did not exceed 1 centimeter in length; large and rounded orbits; absence of contact between the postorbital and jugal bones; markedly shorter forelimbs with a robust digit; elongated

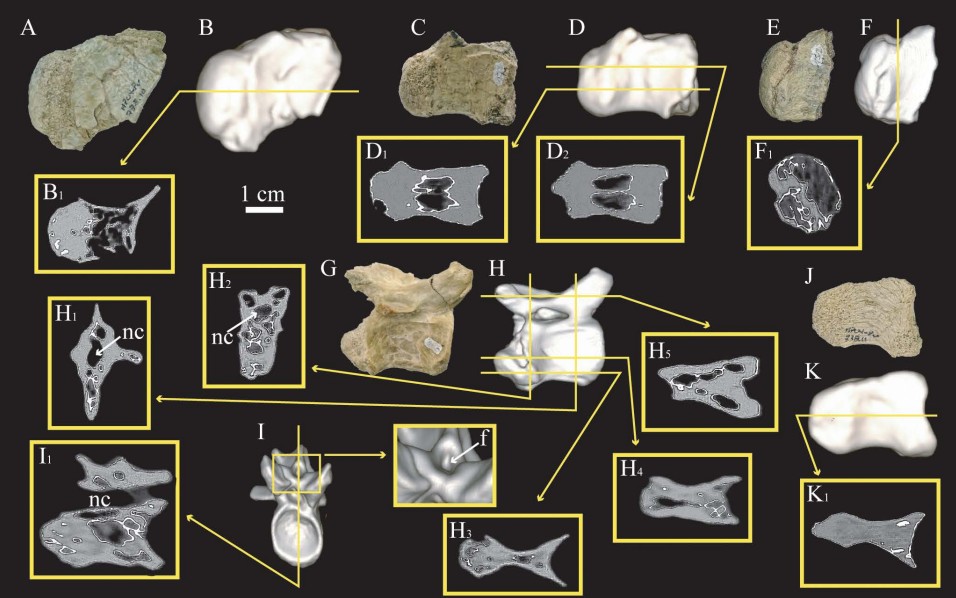

**Fig 7. Caudal vertebrae assigned to cf. *Bonapartenykus ultimus*.** A MPCN-Pv 738.10 in lateral view; B, three-dimensional reconstruction of MPCN-Pv 738.10 and frontal section ($B_1$); **C** MPCN-Pv 738.29 in lateral view; D, three-dimensional reconstruction of MPCN-Pv 738.29 and frontal sections ($D_1$-$D_2$); E, MPCN-Pv 738.30 in lateral view; F, thrsee-dimensional reconstruction of MPCN-Pv 738.30 and transverse section ($F_1$); G, MPCN-Pv 738.8 in lateral view; H, three-dimensional reconstruction of MPCN-Pv 738.8 in lateral view, and transverse sections ($H_1$-$H_2$), and frontal sections ($H_4$-$H_6$); I, three-dimensional reconstruction of MPCN-Pv 738.8 in anterior view, and parasagittal section ($I_1$); J, MPCN-Pv 738.11 in lateral view; K, three-dimensional reconstruction of MPCN-Pv 738.11, and frontal section ($K_1$). Abbreviation: **nc**, neural canal.

hindlimbs; and hyper-elongated tails with procoelous caudal vertebrae, among other features. These morphological features have motivated a number of important studies about the paleobiology of this group [e.g., 5,10,39–44]. Despite this range of past studies, the pneumatic system of the postcranial skeleton has never been investigated in detail. To address this knowledge gap, this study has presented the first contribution focusing on the pneumatic structures of an alvarezsaurid theropod (i.e., *Bonapartenykus*) using first-hand study and CT images.

It has been established that the presence of cortical foramina may be linked to pneumatic diverticula or vasculature [17,20]. However, the persistence of cortical foramina communicating with internal cavities (e.g., camerate tissue) in the axial skeleton is a direct correlate of the presence of pneumatic diverticula [22]. Thus, irrespective of ontogenetic stage, the specimens studied here show unambiguous evidence of postcranial pneumaticity. This study underscores the importance of using CT images in pneumaticity studies, especially in small-bodied taxa in which pneumatic and vascular traces can be difficult to distinguish. For example, the cortical foramina of distal caudal vertebra MPCN-Pv 738.8 was found to be unexpectedly associated with a solid centrum, reinforcing the idea that inferences of pneumaticity should not be based on external osteological correlates alone. Thus, we propose the creation of a more substantial alvarezsaurian CT dataset in the future in order to better assess the distribution and significance of pneumatic features across the group and across theropods more broadly.

The CT images of the presacral vertebrae of *Bonapartenykus ultimus* show, in general terms, a pattern of internal air-spaces. Pneumaticity of cervical and anterior dorsal vertebrae is widespread in theropods [25]. In this aspect, presacral vertebrae of MPCN-PV 738 show camellate internal patterns described for most other coelurosaurs, including early branching

paravians [28,45] and even the early diverging alvarezsaurian *Shishugounykus inexpectus* [33]. The particular case of the dorsal vertebra of the holotype specimen of *B. ultimus* (MPCA 1290) shows no signs of unambiguous pneumaticity; but, observations of this specimen are currently limited to external anatomy only.

Furthermore, specimens studied here show that the sacral region of *Bonapartenykus* was pneumatized, as in at least some abelisauroids and several tetanurans such as spinosauroids, allosauroids, tyrannosauroids, ornithomimids, and maniraptorans including alvarezsaurians [25]. Conversely, apneumatic sacral vertebrae are found in some theropods, such as the carcharodontosaurid *Tyrannotitan* [46] and the therizinosaurid *Nothronychus* [47, 48]; and, although rarely, in some birds too [21].

Caudal pneumaticity has been widely recorded among theropods such as Carcharodontosauridae, Megaraptora, Ornithomimosauria, Therizinosauroidea, Oviraptorosauria, and early diverging alvarezsaurians [17,33,34,48]. Although several clades exhibit caudal pneumaticity, the specimens studied here show that the caudal series of *Bonapartenykus* (though incomplete and represented by disarticulated vertebrae) was pneumatized in both anterior and middle sections. In other words, the pneumatic structures extend into the middle section of the tail. This feature is reminiscent of some ornithomimosaurians whose pneumaticity reached the mid-caudal vertebrae [32]. By contrast, caudal pneumaticity in theropods like carcharodontosaurids [17], therizinosaurs [49, 50], and abelisaurids are typically limited to the anterior caudals, while noasaurids exhibit apneumatic caudal vertebrae [35,51]. Lastly, in oviraptorosaurs, caudal pneumatization reached the posterior caudal vertebrae [35,52,53]. The evidence collected so far shows that the caudal region of the alvarezsaurids had well-developed pneumaticity. Of particular interest is a caudal vertebra of a parvicursorine from the Hell Creek Formation (LACM 153311) with an extensive development of pneumatic structures. The external features of this vertebral element possibly show a greater degree of pneumaticity than South American forms, including *Bonapartenykus*.

The study of axial pneumaticity within Alvarezsauria is limited to external anatomy (such as cortical foramina and natural fractures) and CT scans performed on *Shishugounykus* [33] and now extending to specimens assigned to *Bonapartenykus*. It appears that the earlier-diverging members of Alvarezsauria, such as *Shishugounykus inexpectus*, exhibit a greater degree of pneumatic invasion compared to more derived forms. However, the absence of cortical foramina in this Asiatic form makes the internal spaces of the vertebrae ambiguous evidence of postcranial pneumaticity. Thus, this study based on specimens of *Bonapartenykus* presents the first unambiguous evidence of postcranial pneumaticity in an alvarezsaurian theropod.

The axial skeletons of alvarezsaurians possess internal chambers, in some cases accompanied by cortical foramina that confirm the presence of postcranial pneumaticity. In addition, *Bannykus*, *Xiyunykus* and alvarezsaurids share the presence of a foramen that develops above the neural canal in caudal vertebrae. CT images show that, at least in *Bonapartenykus*, this foramen penetrates the neural arch as unambiguous evidence of alvarezsaurian axial pneumaticity, a feature that requires future work to evaluate among other alvarezsaurian taxa. Nevertheless, what we currently know from alvarezsaurians does not suggest a linear evolutionary trajectory, but rather points to a degree of seemingly random variability in the pneumatic features of the axial skeleton (Fig. 8). Future studies, particularly CT-based analyses of additional alvarezsaurid specimens, will be essential to fully understand the extent of pneumatic invasion among members of this clade and its macroevolutionary significance.

The examined specimens referred to *Bonapartenykus* show a wide range of sizes and ontogenetic stages, indicating a diversity of represented individuals [14]. For instance, cervical neural arch MPCN-Pv 738.28 has rough neurocentral sutures that suggest a lack of fusion to its corresponding vertebral centrum, interpreted as a sign of immaturity. Additionally, the

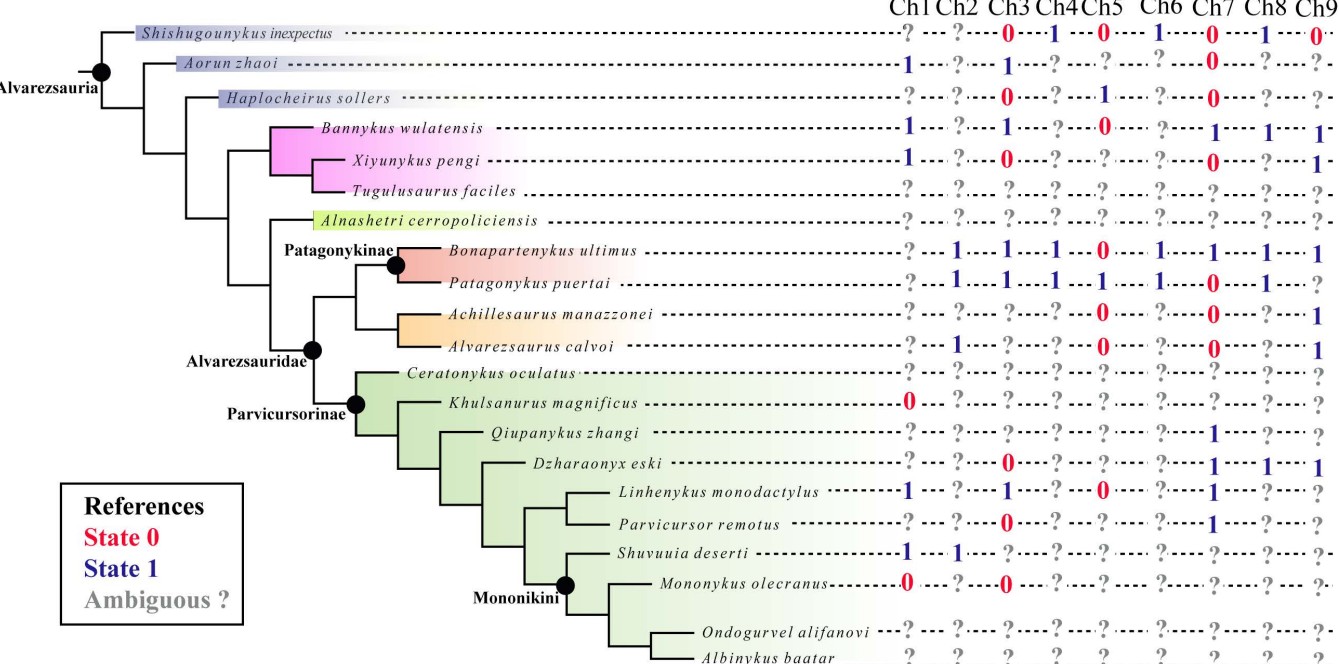

**Fig 8. Mapped pneumatic features in the studies by Meso et al. [ 14].** The studied features are as follows: lateral surface of cervical centra (Ch 1; 0, simple depression; 1, foramina); internal anatomy of cervical centra (Ch 2; 0, absent; 1, present); lateral surface of dorsal centra (Ch 3; 0, simple depression; 1, foramina); internal anatomy of dorsal centra (Ch 4; 0, absent; 1, present); lateral surface of sacral centra (Ch 5; 0, simple depression; 1, foramina); internal anatomy of sacral centra (Ch 6; 0, absent; 1, present); lateral surface of caudal centra (Ch 7; 0, simple depression; 1, foramina); internal anatomy of caudal centra (Ch 8; 0, absent; 1, present); foramen developing above the neural canal in caudal vertebrae (Ch 9; 0, absent; 1, present).

degree of pneumatic invasion within the vertebrae varies considerably between different specimens (possibly linked to ontogenetic stages) particularly in regions such as the tail.

The described specimens have a seemingly random and asymmetrical distribution of foramina and internal pneumatic patterns (as shown by CT scans) along the axial skeleton. This lack of symmetry aligns with the idea proposed by Witmer [54] that the pneumatization process may have been largely opportunistic, potentially stemming from the emergence of pneumatic diverticula due to blood vessels, as suggested by Taylor and Wedel [24]. This random or chaotic internal pneumatization is reminiscent of the internal structure of some sauropods, e.g., a rebbachisaurid caudal vertebra described by Windholz et al [55]. Pneumaticity is variable because pneumatic diverticula followed blood vessels as they developed, and blood vessels are already inherently variable. The nature of pneumatization adds a second layer of variability on top of the variability imposed by the blood vessels. In other words, if blood vessels typically show some degree of variability, then we should expect that any pneumatic systems built on a pattern established by the blood vessels would be even more variable because the process of pneumatization is opportunistic. Thus, the highly variable morphology of the internal air spaces in *Bonapartenykus* is another line of evidence that they are pneumatic rather than vascular.

The occurrence of pneumatic structures in cervical vertebrae and most anterior dorsal vertebrae supports the presence of cervical air-sac diverticula in *Bonapartenykus*. Furthermore, pneumatic chambers in the neural arch of posterior cervical vertebra MPCN-Pv 738.26 connect to the neural canal which supports the presence of paramedullary diverticula, like those of birds [56]. Similarly, sacral and caudal pneumaticity indicate the presence of abdominal

air-sac diverticula. Unfortunately, it is not possible to elucidate the presence of other air-sacs because they do not leave osteological correlates (e.g., caudal thoracic air-sac), or they pneumatize elements not preserved among the specimens here studied (e.g., sternum, sternal ribs) [25]. Finally, while reduced bone mass is advantageous for flight in modern birds, its adaptive significance in non-avian dinosaurs, including theropods of varying sizes, remains uncertain [32]. This highlights the need for further studies of pneumaticity in the different theropod lineages, as well as periodic global reviews [e.g., 31].

## Conclusions

This paper presents the first research focusing on the pneumatic features of an alvarezsaurid theropod based on specimens of *Bonapartenykus ultimus*. It can now be noted that the axial skeleton of this species was pneumatized, and possibly in other alvarezsaurids as well with future work. This premise is supported by the presence of cortical openings communicating with internal air spaces in the vertebral elements of *B. ultimus*. The axial skeleton of *B. ultimus* appears to be pneumatized from the neck to the mid-tail region. Members of Alvarezsauridae do not seem to show a linear evolutionary trend, but rather show a degree of random variability in the pneumatic structures of the axial series. Our findings emphasize the need for studies of the pneumatic system to examine both the external and internal anatomy of specimens. As a priority of future pneumaticity research, specimens should be imaged using available technologies such as CT scanning to fully understand the extent and significance of pneumatic invasion, among alvarezsaurids and theropods more generally.

## Acknowledgments

We thank P. Chafrat from Museo Patagónico de Ciencias Naturales, General Roca, Río Negro Province, Argentina. The authors gratefully acknowledge "Fundacion Patagonica de Ciencias Naturales" and "Sanatorio Juan XXIII" for making the CT images possible. MP was supported by the Faculty of Science of The Chinese University of Hong Kong. We thank Hans-Dieter Sues, an anonymous reviewer, and the editorial team of PLOS ONE for their comments which improved the quality of this manuscript.

## Author contributions

**Conceptualization:** G.J. Windholz, J.G. Meso, M. Pittman.

**Data curation:** G.J. Windholz, J.G. Meso, M. Pittman.

**Formal analysis:** G.J. Windholz, J.G. Meso, M.J. Wedel, M. Pittman.

**Funding acquisition:** G.J. Windholz, J.G. Meso, M. Pittman.

**Investigation:** G.J. Windholz, J.G. Meso, M.J. Wedel, M. Pittman.

**Methodology:** G.J. Windholz, J.G. Meso, M. Pittman.

**Project administration:** G.J. Windholz, J.G. Meso, M. Pittman.

**Resources:** G.J. Windholz, J.G. Meso, M. Pittman.

**Software:** G.J. Windholz, J.G. Meso, M. Pittman.

**Supervision:** G.J. Windholz, J.G. Meso, M. Pittman.

**Validation:** G.J. Windholz, J.G. Meso, M.J. Wedel, M. Pittman.

**Visualization:** G.J. Windholz, J.G. Meso, M.J. Wedel, M. Pittman.

**Writing – original draft:** G.J. Windholz, J.G. Meso, M.J. Wedel, M. Pittman.

**Writing – review & editing:** G.J. Windholz, J.G. Meso, M.J. Wedel, M. Pittman.

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
