## [Decision Letter · Decision Letter 0]

9 Dec 2024

PONE-D-24-50106First unambiguous record of pneumaticity in the axial skeleton of alvarezsaurians (Theropoda: Coelurosauria)PLOS ONE

Dear Dr. Pittman,

Thank you for submitting your manuscript to PLOS ONE. After careful consideration, we feel that it has merit but does not fully meet PLOS ONE’s publication criteria as it currently stands. Therefore, we invite you to submit a revised version of the manuscript that addresses the points raised during the review process.

 As you see in the attached reviews, both reviewers note that you manuscript needs a number of corrections. Please look at them carefully and fulfill them if possible.

We look forward to receiving your revised manuscript.

Kind regards,

Ulrich Joger

Academic Editor

PLOS ONE

2. In your manuscript, please provide additional information regarding the specimens used in your study. Ensure that you have reported human remain specimen numbers and complete repository information, including museum name and geographic location.

For more information on PLOS ONE's requirements for paleontology and archeology research, see https://journals.plos.org/plosone/s/submission-guidelines#loc-paleontology-and-archaeology-research .

3. We note that your Data Availability Statement is currently as follows: [All relevant data are within the manuscript and its Supporting Information files.] Please confirm at this time whether or not your submission contains all raw data required to replicate the results of your study. Authors must share the “minimal data set” for their submission. PLOS defines the minimal data set to consist of the data required to replicate all study findings reported in the article, as well as related metadata and methods (https://journals.plos.org/plosone/s/data-availability#loc-minimal-data-set-definition). For example, authors should submit the following data: - The values behind the means, standard deviations and other measures reported; - The values used to build graphs; - The points extracted from images for analysis. Authors do not need to submit their entire data set if only a portion of the data was used in the reported study. If your submission does not contain these data, please either upload them as Supporting Information files or deposit them to a stable, public repository and provide us with the relevant URLs, DOIs, or accession numbers. For a list of recommended repositories, please see https://journals.plos.org/plosone/s/recommended-repositories. If there are ethical or legal restrictions on sharing a de-identified data set, please explain them in detail (e.g., data contain potentially sensitive information, data are owned by a third-party organization, etc.) and who has imposed them (e.g., an ethics committee). Please also provide contact information for a data access committee, ethics committee, or other institutional body to which data requests may be sent. If data are owned by a third party, please indicate how others may request data access.

Additional Editor Comments (if provided):

Reviewers' comments:

Reviewer's Responses to Questions

**Comments to the Author**

1. Is the manuscript technically sound, and do the data support the conclusions?

Reviewer #1: Yes

Reviewer #2: Partly

2. Has the statistical analysis been performed appropriately and rigorously? 

Reviewer #1: Yes

Reviewer #2: N/A

3. Have the authors made all data underlying the findings in their manuscript fully available?

Reviewer #1: Yes

Reviewer #2: Yes

4. Is the manuscript presented in an intelligible fashion and written in standard English?

Reviewer #1: Yes

Reviewer #2: No

5. Review Comments to the Author

Reviewer #1: The manuscript is very interesting, dealing with a rather rare group of theropods. To me it seems very important to study the pneumaticity of bones in a broad variety of dinosaur taxa. Only a very few mistakes are marked in the attached PDF file.

Reviewer #2: This well-illustrated manuscript presents interesting new information. Unfortunately, the manuscript is very poorly written, requiring extensive editing. Why did co-authors Pittman and Wedel not edit the manuscript? Please note that most journals require that all authors have read and approve the manuscript prior to submission.

6. PLOS authors have the option to publish the peer review history of their article (what does this mean? ). If published, this will include your full peer review and any attached files.

**Do you want your identity to be public for this peer review?** For information about this choice, including consent withdrawal, please see our Privacy Policy .

Reviewer #1: No

Reviewer #2: **Yes: ** Hans-Dieter Sues

---

## [Author Response · Author response to Decision Letter 1]

4 Feb 2025

Our response to the reviewer comments are (repeated in Cover Letter):

Reviewer #1 comments: The manuscript is very interesting, dealing with a rather rare group of theropods. To me it seems very important to study the pneumaticity of bones in a broad variety of dinosaur taxa. Only a very few mistakes are marked in the attached PDF file.

Author reply: All of Reviewer #1’s comments have been adopted.

Reviewer #2 comments: This well-illustrated manuscript presents interesting new information. Unfortunately, the manuscript is very poorly written, requiring extensive editing. Why did co-authors Pittman and Wedel not edit the manuscript? Please note that most journals require that all authors have read and approve the manuscript prior to submission.

Author reply: The manuscript has been carefully revised according to Reviewer #2’s comments and suggestions. The revised manuscript has addressed typographical and grammatical errors and has been proofread by two native English-speakers (MP and MW).

Specific text changes include:

Lines 233-235. “Remarkably, the most complete caudal vertebra MPCN-Pv 738.8 has a large internal air-chamber in the posterior half of the neural arch, above the neural canal, which bifurcates in the anterior half of the neural arch” was replaced by “Remarkably, the most complete caudal vertebra MPCN-Pv 738.8 has a large internal air-chamber in the anterior half of the neural arch, above the neural canal, which bifurcates in the posterior half of the neural arch”

Lines 279-281. “In this aspect, presacral vertebrae of MPCN-PV 738 show camellate internal patterns described for most other coelurosaurs, including early branching avialans such as Rahonavis“ was replaced by “In this aspect, presacral vertebrae of MPCN-PV 738 show camellate internal patterns described for most other coelurosaurs, including early branching paravians”

The final submission comprises 8 Figures and three MS Word files: Revised Manuscript (24 pages), Revised Manuscript with Tracked Changes (25 pages), and this file (Response to Reviewers).

---

## [Editor Report · Decision Letter 1]

8 Nov 2024

PONE-D-24-50106First unambiguous record of pneumaticity in the axial skeleton of alvarezsaurians (Theropoda: Coelurosauria)PLOS ONE

Dear Dr. Pittman,

Thank you for submitting your manuscript to PLOS ONE. After careful consideration, we feel that it has merit but does not fully meet PLOS ONE’s publication criteria as it currently stands. Therefore, we invite you to submit a revised version of the manuscript that addresses the points raised during the review process.

 As you see in the attached reviews, both reviewers note that you manuscript needs a number of corrections. Please look at them carefully and fulfill them if possible.

We look forward to receiving your revised manuscript.

Kind regards,

Ulrich Joger

Academic Editor

PLOS ONE

2. In your manuscript, please provide additional information regarding the specimens used in your study. Ensure that you have reported human remain specimen numbers and complete repository information, including museum name and geographic location.

For more information on PLOS ONE's requirements for paleontology and archeology research, see https://journals.plos.org/plosone/s/submission-guidelines#loc-paleontology-and-archaeology-research .

3. We note that your Data Availability Statement is currently as follows: [All relevant data are within the manuscript and its Supporting Information files.] Please confirm at this time whether or not your submission contains all raw data required to replicate the results of your study. Authors must share the “minimal data set” for their submission. PLOS defines the minimal data set to consist of the data required to replicate all study findings reported in the article, as well as related metadata and methods (https://journals.plos.org/plosone/s/data-availability#loc-minimal-data-set-definition). For example, authors should submit the following data: - The values behind the means, standard deviations and other measures reported; - The values used to build graphs; - The points extracted from images for analysis. Authors do not need to submit their entire data set if only a portion of the data was used in the reported study. If your submission does not contain these data, please either upload them as Supporting Information files or deposit them to a stable, public repository and provide us with the relevant URLs, DOIs, or accession numbers. For a list of recommended repositories, please see https://journals.plos.org/plosone/s/recommended-repositories. If there are ethical or legal restrictions on sharing a de-identified data set, please explain them in detail (e.g., data contain potentially sensitive information, data are owned by a third-party organization, etc.) and who has imposed them (e.g., an ethics committee). Please also provide contact information for a data access committee, ethics committee, or other institutional body to which data requests may be sent. If data are owned by a third party, please indicate how others may request data access.

Additional Editor Comments (if provided):

Reviewers' comments:

Reviewer's Responses to Questions

**Comments to the Author**

1. Is the manuscript technically sound, and do the data support the conclusions?

Reviewer #1: Yes

Reviewer #2: Partly

2. Has the statistical analysis been performed appropriately and rigorously? 

Reviewer #1: Yes

Reviewer #2: N/A

3. Have the authors made all data underlying the findings in their manuscript fully available?

Reviewer #1: Yes

Reviewer #2: Yes

4. Is the manuscript presented in an intelligible fashion and written in standard English?

Reviewer #1: Yes

Reviewer #2: No

5. Review Comments to the Author

Reviewer #1: The manuscript is very interesting, dealing with a rather rare group of theropods. To me it seems very important to study the pneumaticity of bones in a broad variety of dinosaur taxa. Only a very few mistakes are marked in the attached PDF file.

Reviewer #2: This well-illustrated manuscript presents interesting new information. Unfortunately, the manuscript is very poorly written, requiring extensive editing. Why did co-authors Pittman and Wedel not edit the manuscript? Please note that most journals require that all authors have read and approve the manuscript prior to submission.

6. PLOS authors have the option to publish the peer review history of their article (what does this mean? ). If published, this will include your full peer review and any attached files.

**Do you want your identity to be public for this peer review?** For information about this choice, including consent withdrawal, please see our Privacy Policy .

Reviewer #1: No

Reviewer #2: **Yes: ** Hans-Dieter Sues

---

## [Author Response · Author response to Decision Letter 2]

7 Feb 2025

Please find enclosed the revised version of our manuscript ‘First unambiguous record of

pneumaticity in the axial skeleton of alvarezsaurians (Theropoda; Coelurosauria)’ based on

the comments provided by the editor and peer-reviewers. This revised manuscript is now in

journal format. We really appreciate the review feedback provided and believe it has led to

the manuscript improvements that were sought.

Our response to the reviewer comments are:

Reviewer #1 comments: The manuscript is very interesting, dealing with a rather rare group

of theropods. To me it seems very important to study the pneumaticity of bones in a broad

variety of dinosaur taxa. Only a very few mistakes are marked in the attached PDF file.

Author reply: All of Reviewer #1’s comments have been adopted.

Reviewer #2 comments: This well-illustrated manuscript presents interesting new

information. Unfortunately, the manuscript is very poorly written, requiring extensive

editing. Why did co-authors Pittman and Wedel not edit the manuscript? Please note that

most journals require that all authors have read and approve the manuscript prior to

submission.

Author reply: The manuscript has been carefully revised according to Reviewer #2’s

comments and suggestions. The revised manuscript has addressed typographical and

grammatical errors and has been proofread by two native English-speakers (MP and MW).

Specific text changes include:

Lines 233-235. “Remarkably, the most complete caudal vertebra MPCN-Pv 738.8

has a large internal air-chamber in the posterior half of the neural arch, above the

neural canal, which bifurcates in the anterior half of the neural arch” was replaced

by “Remarkably, the most complete caudal vertebra MPCN-Pv 738.8 has a large

internal air-chamber in the anterior half of the neural arch, above the neural canal,

which bifurcates in the posterior half of the neural arch”

Lines 279-281. “In this aspect, presacral vertebrae of MPCN-PV 738 show camellate

internal patterns described for most other coelurosaurs, including early branching

avialans such as Rahonavis“ was replaced by “In this aspect, presacral vertebrae of

MPCN-PV 738 show camellate internal patterns described for most other

coelurosaurs, including early branching paravians”

The final submission comprises 8 Figures and three MS Word files: Revised Manuscript

(24 pages), Revised Manuscript with Tracked Changes (25 pages), and the Response

to Reviewers.

---

## [Editor Report · Decision Letter 2]

14 Feb 2025

First unambiguous record of pneumaticity in the axial skeleton of alvarezsaurians (Theropoda: Coelurosauria)

PONE-D-24-50106R2

Dear Dr. Pittman,

We’re pleased to inform you that your manuscript has been judged scientifically suitable for publication and will be formally accepted for publication once it meets all outstanding technical requirements.

Kind regards,

Felipe Lima Pinheiro, Ph.D

Academic Editor

PLOS ONE

Additional Editor Comments (optional):

During the first round of revisions, the reviewers pointed out only a few minor technical/scientific issues, in addition to noting that the manuscript needed an extensive text revision. It seems to me that this new version addresses all the concerns raised.
---

## [Editor Report · Acceptance letter]

PONE-D-24-50106R2

PLOS ONE

Dear Dr. Pittman,

I'm pleased to inform you that your manuscript has been deemed suitable for publication in PLOS ONE. Congratulations! Your manuscript is now being handed over to our production team.

Kind regards,

on behalf of

Dr. Felipe Lima Pinheiro

Academic Editor

PLOS ONE